# Effects of Combined Remote Ischemic Pre-and Post-Conditioning on Neurologic Complications in Moyamoya Disease Patients Undergoing Superficial Temporal Artery-Middle Cerebral Artery Anastomosis

**DOI:** 10.3390/jcm8050638

**Published:** 2019-05-09

**Authors:** Eun-Su Choi, Yoon-Sook Lee, Byeong-Seon Park, Byung-Gun Kim, Hye-Min Sohn, Young-Tae Jeon

**Affiliations:** 1Department of Anesthesiology and Pain Medicine, Korea University Ansan Hospital, Gyeonggi-do 15355, Korea; potterydoll@gmail.com (E.-S.C.); yslee4719@gmail.com (Y.-S.L.); pbskumc@gmail.com (B.-S.P.); 2Department of Anesthesiology and Pain Medicine, College of Medicine, Inha University Hospital, Incheon 22322, Korea; wangunlove@gmail.com; 3Department of Anesthesiology and Pain Medicine, Ajou university hospital, Suwon-si, Gyeonggi-do 16499, Korea; babysw@naver.com; 4Department of Anesthesiology and Pain Medicine, Seoul National University Bundang Hospital, Seongnam 13620, Korea; 5Department of Anesthesiology and Pain Medicine, Seoul National University College of Medicine, Seoul 03080, Korea

**Keywords:** ischemic postconditioning, ischemic preconditioning, moyamoya disease, surgical anastomosis

## Abstract

Superficial temporal artery-middle cerebral artery (STA-MCA) anastomosis is the most commonly used treatment for Moyamoya disease. During the perioperative period, however, these patients are vulnerable to ischemic injury or hyperperfusion syndrome. This study investigated the ability of combined remote ischemic pre-conditioning (RIPC) and remote ischemic post-conditioning (RIPostC) to reduce the occurrence of major neurologic complications in Moyamoya patients undergoing STA-MCA anastomosis. The 108 patients were randomly assigned to a RIPC with RIPostC group (*n* = 54) or a control group (*n* = 54). Patients in the RIPC with RIPostC group were treated with four cycles of 5-min ischemia and 5-min reperfusion before craniotomy and after STA-MCA anastomosis (RIPostC). The incidence of postoperative neurologic complications and the duration of hospital stay were determined. The overall incidence of neurologic complication was significantly higher in the control group than in the RIPC with RIPostC group (13 vs. 3, *p* = 0.013). The duration of hospital stay was significantly longer in the control group than in the RIPC with RIPostC group (17.8 (11.3) vs. 13.8 (5.9) days, *p* = 0.023). Combined remote ischemic pre- and post-conditioning can be effective in reducing neurologic complications and the duration of hospitalization in Moyamoya patients undergoing STA-MCA anastomosis.

## 1. Introduction

Moyamoya disease (MMD) involves the ends of the intracranial internal carotid arteries bilaterally, and is characterized by progressive stenosis of these vessels and the formation of thin and weak collateral blood vessels like puff of smoke. Direct revascularization surgeries, such as superficial temporal artery-middle cerebral artery (STA-MCA) anastomosis, are the most commonly used treatment for MMD [1,2,3]. However, during surgery and the postoperative period, the brain is exposed to ischemic reperfusion (IR) injury, such that postoperative complications may result in cerebral hypoperfusion or a transient postoperative neurologic deterioration. Hyperperfusion increases the possibility of hemorrhage, which may lead to postoperative morbidity and mortality. Therefore, an effective neuroprotective strategy is needed for patients undergoing STA-MCA anastomosis.

Several animal and human studies have shown that non-lethal ischemic injury has overall protective effects on pre-conditioned tissues and other remote organs, referred to as remote ischemic preconditioning (RIPC) [4,5]. RIPC has cardioprotective effects in patients undergoing cardiac surgery [6]. In terms of neuroprotection, RIPC reduces the recurrence of transient ischemic attack (TIA), improves resilience and increases cerebral perfusion in patients with a history of stroke or TIA, and in those with intracranial arterial stenosis [7]. In patients with subarachnoid hemorrhage, vasodilation protects against ischemia [8]. A brief period of ischemia after IR injury also confers a protective effect on the tissue, a phenomenon known as post-conditioning [9]. Similar to RIPC, post-conditioning protects the heart and brain [9,10,11,12].

The aim of this study was to evaluate the effect of combined RIPC and remote ischemic post-conditioning (RIPostC) on the incidence of neurologic complications in patients undergoing STA-MCA anastomosis.

## 2. Materials and Methods

This study was approved by the Seoul National University Bundang Hospital Institutional Review Board and was registered at ClinicalTrials.gov (NCT03072914). All patients provided informed written consent before enrolment. Patients between the ages of 18 and 65 years with American Society of Anesthesiologists physical status I or II, scheduled for MCA-STA anastomosis under general anesthesia, were enrolled in the study. The exclusion criteria were as follows: history of peripheral vascular arterial or venous diseases; history of peripheral nerve diseases; other brain or cerebrovascular diseases; and/or serious cardiovascular disease, pulmonary disease or kidney disease. The 108 eligible patients were randomly assigned to the RIPC with RIPostC group or the control group using a computer-generated list: an anesthesia nurse not involved in the clinical care of the patients performed the study enrollment using a block randomization list generated by a computer-generated random number sequence.

All patients received standardized anesthetic and surgical techniques during the trial, and all were treated by a single anesthesiologist and three surgeons. Intravenous midazolam (0.03 mg kg^−1^) was administered preoperatively to all patients to relieve anxiety. On arrival at the operating room, monitoring with pulse oximetry, non-invasive blood pressure and electrocardiography was started. The bispectral index (A-2000 BISTM monitor: Aspect Medical Systems, Inc., Natick, MA, USA) was also monitored. Anesthesia was induced and maintained with a target-controlled infusion of remifentanil and propofol (Orchestra infusion pumps; Fresenius Vial, Brezins, France). The dose of remifentanil was adjusted to maintain blood pressure and heart rate within 20% of the preoperative values, and that of propofol was adjusted to maintain a bispectral index of 40–60. After the induction of anesthesia, the patients were administered rocuronium 0.6 mg kg^−1^ and underwent tracheal intubation. The patients’ lungs were mechanically ventilated with oxygen and medical air and an end-tidal carbon dioxide tension of 35–40 mmHg was maintained. Electrocardiography, pulse oximetry, invasive radial artery pressure measurement and oropharyngeal temperature were continuously monitored. Intraoperative neurophysiological monitoring (IONM) tracings were generated by multipulse transcranial electrical stimulation (Xltek Protector; Natus Medical Incorporated Excel-Tech Ltd., Oakville, ON, Canada) at sites 2 cm anterior to the C1 and C2 positions of the electrodes (international 10–20 system) using 3–7 square-wave, monophasic, anodal, constant-voltage electrical pulses of 0.5 millisecond duration, with an interstimulus interval of 2–4 millisecond. IONMs, including motor evoked potentials, were recorded (32-channel Xltek Protector) from the bilateral upper and lower limbs simultaneously, with needles placed bilaterally in the abductor pollicis brevis and abductor digiti minimi, tibialis anterior and abductor hallucis muscles. IONMs were typically recorded every 10 min throughout surgery, but more frequently during critical surgical manipulations. 

In the RIPC with RIPostC group, RIPC commenced after anesthesia induction but before craniotomy, by four cycles consisting of 5 min of ischemia and 5 min of reperfusion (estimated 40 min total) in a lower limb, controlled using a blood pressure cuff inflated to 200 mmHg. Loss of the distal pulse was confirmed by Doppler measurement of the dorsalis pedis pulse. If a pulse was detected, the pressure was increased until it disappeared. After 5 min of ischemia, the cuff was deflated to confirm that the pulse had returned, after which 5 min of reperfusion was initiated. RIPostC was induced just after completion of anastomosis using the same method. In the control group, the same cuff was placed on a lower limb but no pressure was applied. 

Brain computed tomography (CT) and CT angiography, magnetic resonance imaging (MRI), perfusion MRI, Basal brain perfusion single photon emission computed tomography (basal-SPECT) were usually performed before surgery to evaluate the cerebrovascular hemodynamic status. Patients usually underwent CT and CT angiography immediately after MCA-STA anastomosis. In addition, the patients underwent CT angiography 3 days after surgery and brain CT 7 days after surgery, and they were discharged when there were no abnormal findings on postoperative neurologic and radiologic examinations. However, if new neurologic symptoms were developed, brain CT, diffusion-weighted magnetic resonance imaging (MRI), perfusion MRI, and basal-SPECT were performed to detect the cause of neurologic symptoms, such as intracranial hemorrhage, acute cerebral infarction, or cerebral hyperperfusion syndrome.

The primary endpoint was the incidence of major neurologic complications, defined as hyperperfusion syndrome, seizure, epidural hemorrhage, subarachnoid hemorrhage and acute infarction. A neurosurgeon blinded to the group assignment diagnosed and confirmed the major neurologic complications based on clinical symptoms and the change of radiologic images. Detailed descriptions of each neurologic complication are as follows: (i) transient ischemic attack was transient weakness, numbness or paralysis in the face, arm or leg, but there are no abnormalities in Imaging examination. (ii) acute infarction was postoperatively new appeared neurologic symptoms and the ischemic site can be identified by imaging examination. (iii) seizure was movements including violent shaking and a loss of control but there were no abnormalities in imaging examination. (iv) cerebral hyperperfusion syndrome was new development of postoperative focal neurological deficits, neither definite haematomas nor definite acute infarction on a brain CT scan, and a significant focal increase of blood flow at the site of the anastomosis on postoperative single photon emission computed tomography (SPECT). The anesthesiologist who was not involved in the clinical care of the patients performed the ischemic conditioning and another anesthesiologist which does not know the patient’s group collected the outcome. During the perioperative period, the incidence of major adverse events (hyperperfusion syndrome, seizure, epidural hemorrhage, subarachnoid hemorrhage, and acute infarction), mean MCA velocities, and lengths of intensive care unit (ICU) hospital stays were recorded.

The sample size calculation was that used in a previous study, in which the incidence of postoperative neurologic complications was 25%, and was based on a definition of a clinically significant reduction in the incidence of complications of 20% (α = 0.05, power = 0.8). The sample size analysis showed that 54 patients per group would be sufficient to detect a difference between the groups, allowing for a 5% dropout rate. 

The data were expressed as mean ± standard deviation or as numbers. Categorical data were compared using the chi-squared test. Continuous variables were compared using the independent *t*-test, and non-continuous variables using a chi-squared test or Fischer Exact Test. The statistical analyses were performed using SPSS software (ver. 20.0; SPSS Inc., Chicago, IL, USA). A *p* value < 0.05 was considered to indicate statistical significance.

## 3. Results

One hundred and eight patients were randomized and analyzed from March 2017 to July 2018 (Figure 1). Demographic data for the 108 patients enrolled in the study are shown in Table 1. Age, height, weight, body mass index, operation time, anesthetic time, duration of ICU stay, and postoperative MCA velocity were comparable between the two groups (Table 1). However, the duration of hospital stay was significantly shorter in the RIPC with RIPostC group than in the control group (17.8 (11.3) vs. 13.8 (5.9) days, *p* = 0.023) (Table 2).

Although not statistically significant, the postoperative incidence rates of acute infarction, seizure and hyperperfusion syndrome were higher in the control group than in the RIPC with RIPostC group. The overall incidence of neurologic complications was also significantly higher in the control than in the RIPC with RIPostC group (13 vs. 3, *p* = 0.013) (Table 3).

## 4. Discussion

This study showed that as shown in previous studies, while acute infarct did not show statistical significance, acute infarct was more frequent in RIPC with RIPostC group. In addition, due to this difference, there was a significant difference in neurologic complications between the two groups. Also, there was a significant reduction in length of hospital stay in the RIPC with RIPostC group. 

Our results were similar to those of previous studies that evaluated the effect of RIPC on neuroprotection during brain surgery, including on stroke prevention in patients with carotid artery [13] or intracranial [7] arterial stenosis. However, the effect of RIPC in MMD had not been evaluated. In patients undergoing brain tumor surgery [14] and in those with subarachnoid hemorrhage [8,15], RIPC was shown to be effective in reducing the incidence of postoperative ischemic tissue damage. In addition, it had benefits to patients with peripheral vascular disease [16,17].

The mechanism underlying the effect of RIPC on neuroprotection is not clear, although neurovascular action, anti-inflammatory responses, decreased excitotoxicity and metabolic protection have all been implicated [18,19]. In terms of its neurovascular action, neuroprotection by RIPC is thought to be due to the mechanism of nitric oxide (NO) production. When RIPC is performed, No is produced in neuronal nitric oxide synthases (nNOS) by phosphorylated extracellular signal-regulated kinases (p-ERK) activation [20,21,22,23,24]. NO increases cerebral blood flow, improves microvascular perfusion, and maintains brain homeostasis [15,25]. These neuronal pathways are related to neuroprotection, which can be seen more clearly when the endothelial NO synthase disappears and neuroprotection does not occur. The anti-inflammatory response and decreased excitotoxicity are achieved via various intracellular signaling pathways and mediators. The protective effect of reperfusion injury involves the salvage kinase pathway and survivor activating factor enhancement pathway [26,27]. The inflammation is known to play a role in developing cerebral hyperperfusion injury after direct revascularization in patients with Moyamoya disease. Although not statistically significant in our study, the RIPC with RIPostC group showed less seizure or hyperperfusion than the control group. These results are thought to be due to the fact that RIPC induces an anti-inflammatory response, and this effect is thought to be similar to that of RIPC at CRPS patients [28]. RIPC also confers cerebral metabolic protection by decreasing the lactate/pyruvate ratio, an index of cell oxygenation, in patients with aneurysmal bleeding [8]. However, there is a study that local stimulation alone increased the neuroprotective effect of RIPost in the stroke model of rats [29]. Pignataro et al. [23] also showed that neuro-isomers of nitric oxide synthase participate in the neuroprotective effects of RIPost. Thus, neuroprotection can be explained by a systemic response which is similar to cardiac protection and also local responses which are different to cardiac protection. Overall these various mechanisms could explain the shorter hospital stay and reduced neurologic complications of our patients in the RIPC with RIPostC group. 

In contrast to previous studies [8,14,15,16,17], our patients received both RIPC and RIPostC. Most Moyamoya patients have already experienced a TIA or cerebral infarction before surgery. In addition, they are very vulnerable to cerebral infarction during or after surgery due to the surgical manipulations, given that progressive stenosis and formation of thin and weak collateral blood vessels like puff of smoke involves the end of the intracranial internal carotid artery [1,2,3]. Since not only acute ischemic complications but also IR injury or hyperperfusion syndrome are common in Moyamoya patients undergoing STA-MCA anastomosis, during the first few days after surgery preconditioning alone may not be sufficient to prevent reperfusion injury. The effects of RIPostC include promotion of angiogenesis [30]. In a previous study of patients undergoing off-pump coronary artery bypass graft surgery, combined pre-conditioning and post-conditioning was more powerful than pre-conditioning alone [31].

Ischemic pre-conditioning is carried out using various organs or a limb. In this study, we used the lower limb as the conditioning site, because a previous study [21] showed reduced ischemic preconditioning injury with lower rather than upper limb ischemia. RIPostC based on lower limb ischemia was shown to reduce IR injury in humans. The greater protective effects of RIPC and RIPostC conferred by lower limb versus upper limb ischemia may be due to the greater muscle mass of the lower limb, and thus augmented the release of humoral factors.

Several strategies can reduce complications and improve the outcomes of patients after MCA-STA anastomosis surgery. A high blood pressure should be maintained in the awake patient [32] to prevent cerebral vasoconstriction by normocarbia [33,34] and correct anemia, by providing an adequate amount of oxygen [35,36]. These methods were used in our study. Our study additionally demonstrated the potential of RIPC with RIPostC to reduce neurologic complications and improve surgical outcomes.

RIPC has been used for more than a decade, but a standard protocol has yet to be established [37,38]. In the most commonly used protocol, three or four cycles of 5 min ischemia and 5 min reperfusion are administered. In addition, the optimal timing and interval of the stimulus are unclear. Although the duration will differ depending on the primary outcome, the stimulus interval reported among previous studies is extremely variable, with reperfusion applied as soon as immediately before or after the ischemic event, to up to 1 year later [7,16]. The protocol used in our study is the most widely applied and proved to be effective in improving the outcome of patients with MMD. 

Our study had several limitations. First, two conditionings were applied, but their individual effects were not determined. Thus, individual effects of pre- and post-conditioning should be investigated. Second, neurologic complications were analyzed together, rather than separately, due to the low incidence of adverse events. If there was a large number of sample sizes, there would have been a significant difference. Third, in this study, propofol was used to maintain anesthesia. This can affect the protection effect of RIPC. In recent RIPC study, RIPC has been reported to exert cardioprotective effects only under isoflurane anesthesia [39]. It was assumed that propofol, an oxygen free radical scavenger, interfered with the effect of RIPC. However, other studies have also shown that RIPC has been shown to be effective when propofol is used [40], and some studies have not been shown when volatile anesthetics is used [41,42]. We chose propofol because it has advantages over volatile anesthetic in brain surgery. The dose we used in this study was similar to that of a previous study (0.07–0.15 mg/kg/min) [43]. Through previous studies, we cannot deny the possibility that profopol attenuated the effect of RIPC. Because of this, in this study, the effects of RIPC were not clearly demonstrated in terms of postoperative acute ischemia and the effects of RIPC were only shown in terms of hospitalization period. If volatile anesthetics were used, we think the effect of RIPC would have been more obvious. Finally, unfortunately, we did not monitor the entire brain perfusion, including cerebral oximetry. 

## 5. Conclusions

In conclusion, this study demonstrated the potential of combined ischemic conditioning to reduce neurologic complications and the duration of hospitalization in patients undergoing STA-MCA anastomosis for MMD. The results of this study warrant a large multi-center and randomized trial to confirm the efficacy of remote ischemic conditioning.

## Figures and Tables

**Figure 1 jcm-08-00638-f001:**
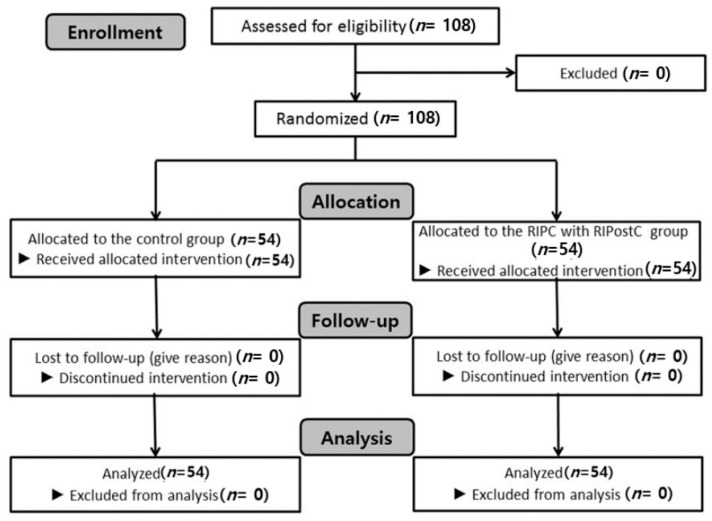
Flow chart of patient enrollment.

**Table 1 jcm-08-00638-t001:** Demographic data.

Demographic Data	Control Group (*n* = 54)	RIPC with RIPostC Group (*n* = 54)	*p* Value
Age (year)	39.0 ± 10.7	37.2 ± 10.8	0.378
Sex (male/female)	20/34 (37.0%/63.0%)	19/35 (35.2%/64.8%)	1.000
Height (cm)	164.7 ± 6.7	164.6 ± 8.0	0.955
Weight (kg)	68.3 ± 14.7	68.5 ± 13.7	0.917
BMI	24.6 ± 4.6	24.7 ± 4.2	0.862
HTN	15 (27.8%)	15 (27.8%)	1.000
DM	1 (1.9%)	4 (7.4%)	0.243
Operation site (right/left)	31/23 (57.4%/42.6%)	26/28 (48.1%/51.9%)	0.441
Operation time (min)	367.7 ± 94.1	360.0 ± 65.6	0.623
Anesthetic time (min)	429.7 ± 106.3	436.0 ± 71.7	0.719

RIPC, remote ischemic pre-conditioning; RIPostC, remote ischemic post-conditioning; BMI, body mass index; min, minutes; HTN, Hypertension; DM, Diabetes mellitus.

**Table 2 jcm-08-00638-t002:** Duration of hospital stay and postanastomosis MCA velocity.

Postoperative Course	Control Group (*n* = 54)	RIPC with RIPostC Group (*n* = 54)	*p* Value
ICU stay duration (day)	2.4 ± 1.0	2.1 ± 0.7	0.092
Hospital stay duration (day)	17.8 ± 11.3	13.8 ± 5.9	0.023
MCA velocity (cc/min)	33.9 ± 22.6	34.6 ± 19.0	0.872

MCA, middle cerebral artery; ICU, intensive care unit; min, minutes.

**Table 3 jcm-08-00638-t003:** Neurologic outcome.

Neurologic Outcome	Control Group (*n* = 54)	RIPC with RIPostC Group (*n* = 54)	*p* Value
Hypoperfusion complication			
TIA	28 (51.9%)	25 (46.3%)	0.700
Acute infarction	8 (14.8%)	2 (3.7%)	0.093
Hyperperfusion complication			
Seizure	3 (5.6%)	0 (0%)	0.243
Hyperperfusion syndrome	2 (3.7%)	1 (1.9%)	1.00
Overall neurologic complication	13 (24%)	3 (5.6%)	0.013

TIA, transient ischemic attack.

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
