# Peer review of "Effects of Combined Remote Ischemic Pre-and Post-Conditioning on Neurologic Complications in Moyamoya Disease Patients Undergoing Superficial Temporal Artery-Middle Cerebral Artery Anastomosis"

_jcm, 2019, doi:10.3390/jcm8050638_

Round 1
Reviewer 1 Report
"The authors present a randomised controlled study regarding the application of RIC in MoyaMoya. A total of 108 patients was recruited.
The results showed a superiority for the RIC group.
The manuscript is of interest to the readership and the methodology is sound.
However, several issues should be adressed.
First, how long was the enrollement period? If i understand correctly, followup ended with discharge?
How was cerebral hyperperfusion syndrome diagnosed? Could the effect of RIC be similar to the response found in inflammatory situations? maybe see pmid: 28340289
Was brain perfusion measured during or after RIC?
Why was the lower limb chosen? How were the cycles chosen?
The patients underwent propofol based anasthesia.
As reported in RIPHeart (NEJM), propofol seems to overwrite or inhibit the RIC effect.
How do the authors explain that cerebral protection is not affected by it?
In the same vein, most work has been done on cardioprotection and stroke. Differences and common ground between heart and brain protection as well as inflammation could be added to the discussion, maybe see PMID 29858664
Also, one should give patients comorbidities and medication, especially DM / AHT and the respective drugs.
Regarding table 3, please give all neurological complications in detail and/or give them under results."
Author Response
Dear editors and reviews
Thank you for reviewing our paper and for giving us precious opinions.
I hope that our answer will be satisfactory to you.
The answers to each question are as follows.
The order of the questions was in the order of reviewer 1, 2.
In addition, we have fixed some spelling and corrected the author's e-mail address.
We have made it possible to recognize the modified place by highlighting them.
Please let us know if you have any further changes.
Best regard
1.The enrollement period was from the day of surgery to the day of discharge.
2. A neurosurgeon, who is blinded to this study, diagnosed cerebral hyperperfusion syndrome. Cerebral hyperperfusion syndrome is diagnosed, as described in the text, by new development of postoperative focal neurological deficits, neither definite haematomas nor definite acute infarction on a brain CT scan, and significant focal increase of blood flow at the site of the anastomosis on postoperative SPECT. Cerebral hyperperfusion syndrome after STA-MCA anastomosis results from a rapid increase in cerebral blood flow in the chronic ischemic brain. The inflammation is known to play a role in developing cerebral hyperperfusion injury after direct revascularization in patients with moyamoya disease. Although not statistically significant in our study, the RIPC with RIPostC group showed less seizure or hyperperfusion than the control group. These results are thought to be due to the fact that RIPC induces an anti-inflammatory response, and this effect is thought to be similar to that of RIPC at pmid: 28340289. we added this to the discussion line 191-196.
3. Unfortunately, we did not monitor the entire brain perfusion, such as cerebraloximetry. However, the perfusion status was monitored by measuring perfusion MRI, basal-SPECT, and middle cerebral artery velocity after surgery. We add this to the limitation section in discussion line 249-250.
4. Why was the lower limb chosen? How were the cycles chosen? ->In this study, as described in the text, we used the lower limb as the conditioning site, because a previous study showed reduced ischaemic preconditioning injury with lower rather than upper limb ischaemia. In reference 6, most studies performed 3-4 RIPCs. We thought that it would be enough to perform 4 times rather than 3 times and we performed 4 times.
5. The patients underwent propofol based anasthesia. As reported in RIPHeart (NEJM), propofol seems to overwrite or inhibit the RIC effect. How do the authors explain that cerebral protection is not affected by it?
->In this study, propofol was used to maintain anesthesia. This can affect the protection effect of RIPC. In recent RIPC studies, RIPC has been reported to exert cardioprotective effects only under isoflurane anesthesia. It was assumed that propofol, an oxygen free radical scavenger, interfered with the protective effect of RIPC. However, other studies have also shown that RIPC has been shown to be effective when propofol is used, and some studies have not been shown when volatile anesthetics is used. We chose propofol because it has advantages over volatile anesthetic in brain surgery. The dose we used in this study was similar to that of previous study (0.07-0.15 mg / kg / min). Through previous studies, we can not deny the possibility that profopol atteneuated the effect of RIPC. And because of this, in this study, the effects of RIPC were not clearly demonstrated in terms of postoperative acute ischemia and the effects of RIPC were only showed in terms of hospitalization period. If volatile anesthetics were used, we think the effect of RIPC would have been t be more obvious. we added this to the limitation section in discussion line 238-249.
6. In the same vein, most work has been done on cardioprotection and stroke. Differences and common ground between heart and brain protection as well as inflammation could be added to the discussion, maybe see PMID 29858664
->We have added to the discussion the similarities and differences of heart and brain protection in discussion, line 198-201.
7. Also, one should give patients comorbidities and medication, especially DM / AHT and the respective drugs.
->We have added in table 1
8. Regarding table 3, please give all neurological complications in detail and/or give them under results."
A detailed description of each neurologic complication is given below and added to the method section, line 122-129.
Transient ischemic attacks – transient weakness, numbness or paralysis in face, arm or leg, but there are no abnormalities in Imaging examination
Acute infacrtion – postoperatively new appeared neurologic symptom and the ischemic site can be identified by imaging examination.
Seizure - complex partial seizure but there are no abnormalities in Imaging examination
Cerebral hyperperfusion syndrome - new development of postoperative focal neurological deficits, neither definite haematomas nor definite acute infarction on a brain CT scan, and significant focal increase of blood flow at the site of the anastomosis on postoperative SPECT
9.Only concerns are on the Discussione of previously published paper. For istance, the putative role of NO as mechanism of protection is only slightly discussed and some seminal works are not mentioned, i.e. Pignataro G et al., Neurobiol of Disease, 2013; Hess D et al., Stroke, 2013. -> The putative role of NO as mechanism of protection has been added in more detail in discussion section line 182-188. .
Reviewer 2 Report
The paper by Choi et al supports the hypothesis that combined remote ischaemic pre- and post conditioning can be effective in reducing neurologic complications and the duration of hospitalisation in moyamoya patients undergoing STA-MCA anastomosis.
The paper is interesting and the results are supported by experimental data.
Only concerns are on the Discussione of previously published paper. For istance, the putative role of NO as mechanism of protection is only slightly discussed and some seminal works are not mentioned, i.e. Pignataro G et al., Neurobiol of Disease, 2013; Hess D et al., Stroke, 2013.
Author Response
Dear editors and reviews
Thank you for reviewing our paper and for giving us precious opinions.
I hope that our answer will be satisfactory to you.
The answers to each question are as follows.
The order of the questions was in the order of reviewer 1, 2.
In addition, we have fixed some spelling and corrected the author's e-mail address.
We have made it possible to recognize the modified place by highlighting them.
Please let us know if you have any further changes.
Best regard
1.The enrollement period was from the day of surgery to the day of discharge.
2. A neurosurgeon, who is blinded to this study, diagnosed cerebral hyperperfusion syndrome. Cerebral hyperperfusion syndrome is diagnosed, as described in the text, by new development of postoperative focal neurological deficits, neither definite haematomas nor definite acute infarction on a brain CT scan, and significant focal increase of blood flow at the site of the anastomosis on postoperative SPECT. Cerebral hyperperfusion syndrome after STA-MCA anastomosis results from a rapid increase in cerebral blood flow in the chronic ischemic brain. The inflammation is known to play a role in developing cerebral hyperperfusion injury after direct revascularization in patients with moyamoya disease. Although not statistically significant in our study, the RIPC with RIPostC group showed less seizure or hyperperfusion than the control group. These results are thought to be due to the fact that RIPC induces an anti-inflammatory response, and this effect is thought to be similar to that of RIPC at pmid: 28340289. we added this to the discussion line 191-196.
3. Unfortunately, we did not monitor the entire brain perfusion, such as cerebraloximetry. However, the perfusion status was monitored by measuring perfusion MRI, basal-SPECT, and middle cerebral artery velocity after surgery. We add this to the limitation section in discussion line 249-250.
4. Why was the lower limb chosen? How were the cycles chosen?
->In this study, as described in the text, we used the lower limb as the conditioning site, because a previous study showed reduced ischaemic preconditioning injury with lower rather than upper limb ischaemia. In reference 6, most studies performed 3-4 RIPCs. We thought that it would be enough to perform 4 times rather than 3 times and we performed 4 times.
5. The patients underwent propofol based anasthesia. As reported in RIPHeart (NEJM), propofol seems to overwrite or inhibit the RIC effect. How do the authors explain that cerebral protection is not affected by it?
->In this study, propofol was used to maintain anesthesia. This can affect the protection effect of RIPC. In recent RIPC studies, RIPC has been reported to exert cardioprotective effects only under isoflurane anesthesia. It was assumed that propofol, an oxygen free radical scavenger, interfered with the protective effect of RIPC. However, other studies have also shown that RIPC has been shown to be effective when propofol is used, and some studies have not been shown when volatile anesthetics is used. We chose propofol because it has advantages over volatile anesthetic in brain surgery. The dose we used in this study was similar to that of previous study (0.07-0.15 mg / kg / min). Through previous studies, we can not deny the possibility that profopol atteneuated the effect of RIPC. And because of this, in this study, the effects of RIPC were not clearly demonstrated in terms of postoperative acute ischemia and the effects of RIPC were only showed in terms of hospitalization period. If volatile anesthetics were used, we think the effect of RIPC would have been t be more obvious. we added this to the limitation section in discussion line 238-249.
6. In the same vein, most work has been done on cardioprotection and stroke. Differences and common ground between heart and brain protection as well as inflammation could be added to the discussion, maybe see PMID 29858664
->We have added to the discussion the similarities and differences of heart and brain protection in discussion, line 198-201.
7. Also, one should give patients comorbidities and medication, especially DM / AHT and the respective drugs.
->We have added in table 1
8. Regarding table 3, please give all neurological complications in detail and/or give them under results."
A detailed description of each neurologic complication is given below and added to the method section, line 122-129.
Transient ischemic attacks – transient weakness, numbness or paralysis in face, arm or leg, but there are no abnormalities in Imaging examination
Acute infacrtion – postoperatively new appeared neurologic symptom and the ischemic site can be identified by imaging examination.
Seizure - complex partial seizure but there are no abnormalities in Imaging examination
Cerebral hyperperfusion syndrome - new development of postoperative focal neurological deficits, neither definite haematomas nor definite acute infarction on a brain CT scan, and significant focal increase of blood flow at the site of the anastomosis on postoperative SPECT
9.Only concerns are on the Discussione of previously published paper. For istance, the putative role of NO as mechanism of protection is only slightly discussed and some seminal works are not mentioned, i.e. Pignataro G et al., Neurobiol of Disease, 2013; Hess D et al., Stroke, 2013.
-> The putative role of NO as mechanism of protection has been added in more detail in discussion section line 182-188.
Round 2
Reviewer 1 Report
I approve of the changes made and thank the authors for their thorough discussion of the issues raised.
Author Response
Dear editors and reviewers.
Thank you very much for your valuable feedback so that I can become a better paper.
Some spelling errors and grammar errors have been corrected as follows:
English spelling and American spelling are mixed and amended and spelled out in American spelling.
English spelling and American spelling are mixed and amended and spelled out in American spelling.
And we corrected the grammatical error in line 183, 245, 246, and 249.
(%) Was inserted in the incidence of occurrence in table 1 and 3.
The period of study was inserted in the result part, line 146.
I hope you have satisfied with our revision.
Sincerely,
Eun-su Choi
Department of Anesthesiology and Pain Medicine
Korea University Ansan Hospital, Gyeonggi-do, Republic of Korea
E-mail: [email protected]